# The Human Mutation K237_V238del in a Putative Lipid Binding Motif within the V-ATPase a2 Isoform Suggests a Molecular Mechanism Underlying Cutis Laxa

**DOI:** 10.3390/ijms25042170

**Published:** 2024-02-11

**Authors:** Anh Chu, Yeqi Yao, Miroslawa Glibowicka, Charles M. Deber, Morris F. Manolson

**Affiliations:** 1Faculty of Dentistry, University of Toronto, Toronto M5G 1G6, ON, Canada; anhnt.chu@mail.utoronto.ca (A.C.); yeqiyao@gmail.com (Y.Y.); 2Division of Molecular Medicine, Research Institute, Hospital for Sick Children, Toronto M5G 0A4, ON, Canada; miragl@ymail.com (M.G.); deber@sickkids.ca (C.M.D.); 3Department of Biochemistry, Faculty of Medicine, University of Toronto, Toronto M5S 1A8, ON, Canada

**Keywords:** V-ATPases, V-ATPase a2 isoforms, protein–lipid interaction, phosphoinositides, PI(4)P, cutis laxa

## Abstract

Vacuolar ATPases (V-ATPases), proton pumps composed of 16 subunits, are necessary for a variety of cellular functions. Subunit “a” has four isoforms, a1–a4, each with a distinct cellular location. We identified a phosphoinositide (PIP) interaction motif, KX_n_K(R)IK(R), conserved in all four isoforms, and hypothesize that a/PIP interactions regulate V-ATPase recruitment/retention to different organelles. Among the four isoforms, a2 is enriched on Golgi with a2 mutations in the PIP motif resulting in cutis laxa. We hypothesize that the hydrophilic N-terminal (NT) domain of a2 contains a lipid-binding domain, and mutations in this domain prevent interaction with Golgi-enriched PIPs, resulting in cutis laxa. We recreated the cutis laxa-causing mutation K237_V238del, and a double mutation in the PIP-binding motif, K237A/V238A. Circular dichroism confirmed that there were no protein structure alterations. Pull-down assays with PIP-enriched liposomes revealed that wildtype a2NT preferentially binds phosphatidylinositol 4-phosphate (PI(4)P), while mutants decreased binding to PI(4)P. In HEK293 cells, wildtype a2NT was localized to Golgi and co-purified with microsomal membranes. Mutants reduced Golgi localization and membrane association. Rapamycin depletion of PI(4)P diminished a2NT-Golgi localization. We conclude that a2NT is sufficient for Golgi retention, suggesting the lipid-binding motif is involved in V-ATPase targeting and/or retention. Mutational analyses suggest a molecular mechanism underlying how a2 mutations result in cutis laxa.

## 1. Introduction

Vacuolar H^+^-ATPases (V-ATPases) are conserved ATP-dependent proton pumps responsible for the maintenance of organelle luminal pH in eukaryotic cells [1,2,3,4,5]. They are multi-subunit complexes comprised of a cytosolic V_1_ sector responsible for ATP hydrolysis coupled to proton translocation through a membrane-bound V_o_ sector [6]. Several subunits have tissue- or organelle-specific isoforms thought to mediate V-ATPase localization to their various functional cellular destinations [7,8]. In mammals, there are four isoforms of the V_o_-*a* subunit, a1–a4, which play key roles in V-ATPase localization [9,10,11,12,13]. The a1–a3 isoforms are expressed ubiquitously in different tissues, in which a1 and a2-containing V-ATPases are mainly found on intracellular membranes [14,15,16,17,18,19,20], while a3 are found in both intracellular compartments and the plasma membrane [21,22,23]. The a4 isoform is restricted to the kidney [24,25], epididymis [26], and inner ear [27], where a4-containing V-ATPases are targeted to the plasma membrane. Mutations in “a” subunit isoforms are linked to various diseases. Mutations within a1 have been linked to epileptic encephalopathy [14,16]. The autosomal recessive disease, cutis laxa, is characterized by a glycosylation malfunction in the Golgi and is associated with mutations in the a2 isoform [28,29,30,31]. Mutations in a3 and a4 are linked to osteopetrosis [32,33,34,35] and distal renal tubular acidosis [25,36,37], respectively.

The 90kDa *a*-subunit contains a cytosolic N-terminal domain and a membrane-bound C-terminal domain consisting of eight transmembrane helices (TMs) [10,38,39]. In *Saccharomyces cerevisiae*, the sorting signal for V-ATPases within the N-terminal half of the *a*-subunit homolog, Stv1p, is well characterized [10,11]. The sorting information, W^83^KY, present in the Stv1p isoform dictates the retention of Stv1p-containing V-ATPases within the Golgi network [10]. However, little is known about the targeting signals for the mammalian Golgi-specific a2 isoform. Evidence in yeast suggests the interaction of the *a* subunits with membrane phosphoinositides can account for the membrane retention of the V-ATPases at specific locations [40,41].

Phosphoinositides (PIPs) are generated by reversible phosphorylation of the precursor phosphatidylinositol (PI) at the inositol headgroup, mediated by organelle-specific kinases and phosphatases [42,43,44]. Organelle-specific distribution and composition of the seven different PIPs play important roles in membrane protein trafficking [42,45]. Spatial or temporal enrichment of PIPs at microdomains within organelle membranes is involved in regulating the activity of membrane-bound ion channels and transporters [45,46]. The cytosolic N-terminal domain of the yeast *a* isoform, Vph1p, was recruited to the vacuolar membrane when the level of endosome/vacuole-specific PI(3,5)P_2_ was elevated [41]. In contrast, Stv1p was shown to bind directly to a Golgi-specific PI(4)P, and the interaction was attributed to its cytosolic N-terminal domain [40].

In previous work, we showed that the plasma membrane-specific a4 isoform preferentially binds to PI(4,5)P_2_ and that the a4NT-PI(4,5)P_2_ interaction is responsible for membrane retention [47]. We proposed a putative binding domain within the N-terminal half of the *a* subunit, containing a conserved basic motif (K/R)X(K/R)(K/R). In the present study, we found that the a2 isoform interacts in vitro with PI(4)P, a Golgi-enriched PIP, and show that PI(4)P at the Golgi helps to retain the cytosolic N-terminal domain of a2 at the Golgi membrane. Additionally, we show that the cutis laxa causing mutation K237_V238del within the critical a2 isoform K^237^VKK^240^ binding motif not only reduced interactions with PI(4)P but also disrupted protein membrane retention, suggesting the molecular mechanism underlying the disease. 

## 2. Results

### 2.1. Mutations within the Putative Binding Motif reduced Interaction of a2 with PI(4)P-Enriched Liposomes In Vitro

In our previous work, we proposed a putative lipid binding domain located at the distal lobe of the N-terminal half of a4. We further suggested that a conserved basic motif, (K/R)X(K/R)(K/R), is critical for the interaction with the acidic headgroup of PIPs [47]. In *a*2, we propose this critical lipid binding motif is K^237^VKK^240^ (Figure 1). Using mutagenesis to verify the basic motif, we generated two mutations in the *a*2 N-terminal domain (a2NT), K237A/V238A and K237_V238del, both within the putative binding motif. The K237_V238del is a mutation found within humans that results in cutis laxa [48]. 

We first assessed whether either mutation affected protein folding using circular dichroism (CD) spectroscopy. WT a2NT contains a mainly helical structure, exhibited by two negative minima at 222 nm and 208 nm. The CD spectra of the mutants and WT, all in aqueous buffer, overlap at the characteristic wavelengths, indicating that the mutations did not alter protein structure (Figure 2A). To mimic the membrane environment and determine whether the presence of micelles enhances the helicity of the proteins, we added detergent with an SDS-to-peptide ratio of 370:1. The spectra of WT and mutants behaved similarly in the presence of SDS micelles with increased negative ellipticity at 208 nm and 222 nm, suggesting the membrane-bound behavior of the proteins (Figure 2B). Variation in the signal of the positive peak at 190–200 nm is likely due to different trace amounts of imidazole in the buffer.

We tested in vitro PIP interactions of wildtype and mutants with PIP-enriched liposomes using a liposome pull-down assay. Wildtype and mutants were incubated with PolyPIPosome liposomes enriched with different PIPs, including the Golgi-specific PI(4)P, and its derivatives PI(3,4)P_2_,PI(4,5)P_2_ and PI(3,4,5)P_3_ (labelled as PIP3). The protein–liposome complexes are collected via high-speed centrifugation and resolved by Western blot. WT a2NT showed significantly higher binding to PI(4)P compared to PI(3,4)P_2_ and PIP3 (Figure 3A,B). While both mutations visually appear to reduce association with all polyPIPosome liposomes (Figure 3B left panel), only the liposomes enriched with PI(4)P resulted in a significant reduction (Figure 3B right panel). The binding differences between WT and mutants to I(3,4)P_2_, PI(4,5)P_2_, and PIP3 were not statistically significant (Figure 3C). This result is consistent with the fact that the a2 isoform is localized in Golgi and that PI(4)P is primarily found in Golgi.

These data suggest that the basic motif K^237^VKK^240^ in a2 is required for PI(4)P interaction, and mutations within this motif negatively impact the interaction. This further suggests that disruption of the a2-PI(4)P interaction could be the molecular mechanism underlying the disease-causing mutation, K237_V238del.

### 2.2. Mutations K237A/V238A and K237_V238del Reduce a2NT Golgi Localization

To assess cytosolic a2NT membrane association, we expressed FLAG-tagged wildtype a2NT and mutants a2NT K237A/V238A and K237_V238del in HEK293 cells and performed subcellular fractionation. Although the a*2*NT.FLAG lacks a transmembrane domain, it was still detected in microsomal fractions, suggesting that the membrane-bound C-terminal domain is not essential for membrane retention and that the N-terminal domain is sufficient to bring the protein to the membrane (Figure 4A). There was a significant decrease in the amount of both mutants in the microsomal fraction compared to the wildtype (Figure 4B). These results indicate that both mutations disrupt PI(4)P binding in vitro and reduce membrane retention in vivo.

Immunofluorescence microscopy was used to visualize the localization of wildtype and mutant a2NT in the HEK293 cells. Wildtype a2NT.FLAG (red) were enriched at the Golgi, visualized with the Golgi specific marker, Tgn38-CFP (Figure 5A, top row). As a2NT was recruited to the Golgi in the absence of a membrane-bound C-terminal domain, this suggests that Golgi sorting information lies within the cytosolic N-terminal half. In contrast, there was a significant reduction in the Golgi localization of a2.K237A/V238A (Figure 5A, middle row) and a2.K237_V238del (Figure 5A, bottom row) (Figure 5B). This result aligns with the diminished membrane retention of the two mutants in microsomal fractions, supporting the hypothesis that the PI(4)P binding motif in a2 is, in part, responsible for Golgi membrane targeting/retention. 

### 2.3. Depletion of Golgi PI(4)P Impairs a2NT Recruitment to Golgi

We next tested whether the depletion of Golgi PI(4)P impairs a2NT Golgi recruitment. Sac1 is a PI(4)P phosphatase converting PI(4)P to PI [49,50]. We recruited Sac1 phosphatase to Golgi using the rapamycin-induced dimerization method [51,52,53]. Sac1 phosphatase coupled to FK506 binding protein FKBP (Sac1-FKBP) and Golgi membrane anchor Tgn38 coupled to FKBP-rapamycin binding domain FRB (Tgn38-FRB-CFP) were dimerized by the addition of rapamycin (Figure 6A). Sac1-PJ phosphatase is recruited to the Golgi (Figure 6B), where it converts PI(4)P to PI, resulting in the depletion of the Golgi pool of PI(4)P.

HEK293 cells were co-transfected with plasmids containing a2NT.FLAG, Sac1-PJ-FKBP, and Tgn38-FRB-CFP. a2NT co-localized with Tgn38-CFP, indicating recruitment to the Golgi (Figure 7A, top row). Additionally, 100 nM rapamycin decreased the a2NT intensity at the Golgi (Figure 7A, bottom row), suggesting that depletion of Golgi PI(4)P impairs a2NT’s localization/retention at the Golgi.

## 3. Discussion

The cytosolic N-terminal domain of the *a* subunit serves as a connector for V_1_ and V_o_ assembly [54] and is an important target for multiple V-ATPase regulators [15,17,20,21]. Experiments with chimeras of the two yeast orthologs, Vph1p and Stv1p, showed that the *a*NT contains information for both the localization and regulation of V-ATPase by reversible assembly [13]. Phosphoinositides regulate transmembrane channels and transporters [42,45,46]. Interactions between *a* subunit isoforms and different PIPs may impact both functional regulation and localization, which, in turn, could account for the differences in functional destinations among *a* subunit isoforms.

Previously, we proposed a putative lipid binding domain in the *a* subunit and the possible involvement of this domain in V-ATPase regulation [47]. Sequence alignment of the four mammalian V-ATPase *a* subunit isoforms with the two yeast orthologs revealed a conserved basic motif KX_n_K(R)IK(R) required for the PIP’s interaction in the N-terminal domain of a4. Here, we provide evidence that the Golgi-specific a2 isoform directly interacts with PI(4)P, a Golgi-enriched PIP. Our results with a2 mutations, K237A/V238A and K237_V238del, indicate that mutations within the binding motif (K^237^VKK^240^) compromised the membrane association and Golgi localization, further suggesting that this basic motif is essential for PI(4)P binding. The depletion of Golgi PI(4)P with rapamycin had similar effects on a2 localization, indicating that a2-PI(4)P interaction is important for the Golgi localization of a2-containing V-ATPases. A recent study identified another PI(4)P interaction sequence K^221^WY within the vicinity of the putative binding domain, and the mutation of the K221 residue compromised PI(4)P binding [55]. PIPs are expressed as lipid rafts within the membrane [56,57] so that multiple basic residues are exposed to the membrane, which would strengthen protein–lipid interactions. We hypothesize that the K^221^WY sequence could help to strengthen the PIP/protein interaction or potentially define the PIP’s specificity of the a2 isoform. 

PIP binding is often associated with protein conformational changes [58,59,60,61]. Structural analyses of yeast V-ATPases suggest that there are conformational changes between Vph1pNT in holoenzyme V_1_-V_o_ and in free V_o_, as well as movement of the N-terminal domain in different states of the active enzyme [6,62]. Our putative lipid binding motif is within the distal lobe of the *a*NT, which rotates between the different V-TPase active states. Similar to the stabilization of K^+^ channel Kir2.2 upon binding to PI(4,5)P_2_ [59], it is possible that binding to the Golgi PI(4)P traps the a2 isoform at the Golgi membrane as well as in a conformational state that promotes V-ATPase assembly and/or activity. Studies in yeast indicate that mutations resulting in the loss of PI(4)P binding compromise Stv1p-containing V-ATPases’ function, resulting in growth defects at alkaline pH [40]. Structural studies of conformational changes induced by PIP binding can provide mechanistic insights for the functional role of this interaction. Nevertheless, *a*NT is the target of multiple V-ATPase regulators [15,17,20,41,63], and PIP interaction is only one mode of V-ATPase regulation at specific membranes. 

The a2 K237_V238del mutation has been identified in patients with cutis laxa [48]. Here, we show that this mutation disrupted PI(4)P interaction and compromised Golgi localization. The characterization of conserved residues implicated in diseases has been successfully used to determine functional domains and has informed the discovery of novel therapeutic targets [64,65,66]. Understanding *a*-PIP interactions and their impact on V-ATPase localization and regulation could similarly inform the development of a therapeutic control of V-ATPase subpopulations, enabling the inhibition, specifically, of V-ATPases involved in osteoporosis [7,21], and metastatic cancer [67,68].

## 4. Materials and Methods

### 4.1. Expression and Purification of Human a2NT Wildtype and Mutants K237A/V238A, K237_V238.del from E. coli

pET32a+ :: human V-ATPase a2NT (MM1115). N-terminal domain of human a2 from ATG to T400 was obtained by PCR with primers MO501: 5′-ACGTGGTACCATGGGCTCCATGTTCCGGAG and MO502: 5′-ACGTGAATTCACAGATCTCCGCCGGTGTAGGGAGCGGGGTTGAC. The PCR product was cloned into pET32a(+) plasmids between *KpnI* and *EcoRV* sites, resulting in MM1111. pcDNA3 :: human V-ATPase a2NT (MM1127) *KpnI* and *EcoRV* insert of a2NT in MM1115 was moved into pcDNA3.1+; the new construct was named MM1127. pET32a+ :: human V-ATPase a2NT K237A/V238A (MM1121) Q5 Site-Directed Mutagenesis Kit (NEB E0554S) was used on MM1115 to make human V-ATPase a2NT.K237A/V238A mutant with primers MO523: 5′-GCAACATCGACGTCACCCAGCAG and MO524: 5′-ACGTGAATTCTTAGGTGTAGGGGGCTGGGTTTATC, sequencing verified. pcDNA3 :: human V-ATPase a2NT K237A/V238A (MM1128) *KpnI* and *EcoRV* fragment of MM1121 was ligated into pcDNA3.1+, and the new construct was named MM1128. pET32a+ :: human V-ATPase a2NT K237_V238.del (MM1122) Q5 Site-Directed Mutagenesis Kit (NEB E0554S) was used on MM1115 to make human V-ATPase a2NT.K238_V238.del mutant. pcDNA3 :: human V-ATPase a2NT K237_V238.del (MM1129) The *KpnI* and *EcoRV* insert of MM1122 was ligated into pcDNA3.1+; the new construct was named MM1129. 

The N-terminal domain of human a2 (amino acid 1–400) with a 6X His tag at the carboxyl end was expressed in E.coli Rosetta (DE3) via pET32a plasmid and purified as described in a previous study [47].

### 4.2. PolyPIPosome Pull-Down Assay

In total, 20 µg of purified proteins, a2NT wildtype, K237A/V238A, and K237_V238.del, was incubated with 20 µL 1 mM PolyPIPosomes (Echelon, US) and 200 µL of binding buffer (50 mM Tris pH8.0, 150 mM NaCl, and 0.05% Nonidet P-40). Pull-down assay was performed as in [47] as well as a Western blot with mouse anti-His antibody (Sigma SAB1305538) and goat anti-mouse IgG secondary antibody (Invitrogen 31430). 

### 4.3. HEK293 Transfection and Cellular Fractionation 

HEK293 cells (ATCC, US) were cultured on 10 cm culture dishes in Dulbecco’s modified Eagle’s medium (DMEM) (Gibco, US) containing 10% fetal bovine serum (FBS) and 0.5% antibiotics, and grown in a 95% air, 5% CO_2_ humidified environment at 37 °C. pcDNA3 plasmids of human a2 N-terminal domain (amino acid 1–400) wildtype and mutants K237A/V238A, K237_V238.del, 5 μg of plasmid/dish, were transfected into HEK293 cells using PolyJet Reagent (SignaGen, US) in accordance with the procedure recommended by the manufacturer. Cellular fractionation was as described in [47]. The fractions were analyzed by Western blot with anti-atp6v0a2 (Abcam, UK ab96803).

### 4.4. Immunofluorescence 

In the rapamycin treatment experiment, cells were treated with 100 nM of rapamycin for 15 min at room temperature before fixing. Images were acquired with a confocal microscope (Leica Confocal SP8, Germany) using a 63x oil objective. Colocalizations were measured with Mander’s coefficient M1 [69]. Antibodies used were as follows: mouse anti-DDDDK tag (Abcam ab18230), goat anti-mouse IgG Alexa Fluor ^TM^ 647 (Invitrogen A21235), and DAPI stain (Invitrogen D1306). 

### 4.5. Statistical Analysis

GraphPad Prism 9.4.1 software was used for statistical analysis and statistical graph production. One-way ANOVA followed by Dunnett’s multiple comparison test or Student’s *t*-test were used as indicated in figure legends. In figures, asterisks are used as follows: * indicates *p* < 0.05, ** indicates *p* < 0.01, and *** indicates *p* < 0.001. The experimental results are expressed as the mean  ±  SEM.

## Figures and Tables

**Figure 1 ijms-25-02170-f001:**
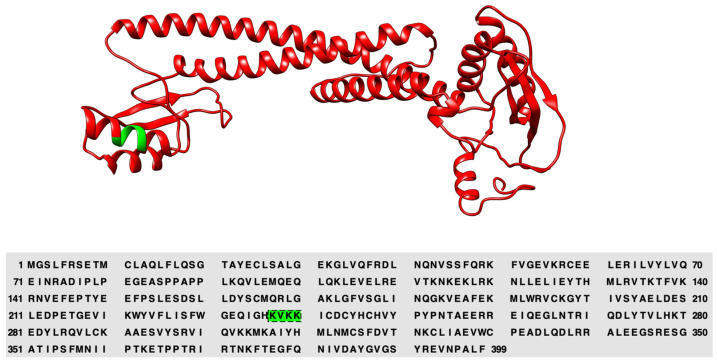
a2NT model generated by Phyre2.0 with the critical basic K^237^VKK^240^ motif in the distal lobe highlighted in green.

**Figure 2 ijms-25-02170-f002:**
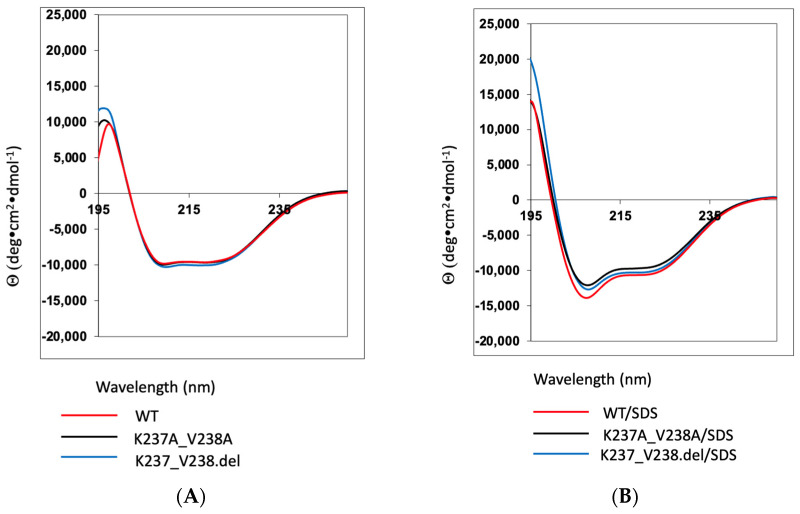
Mutations K237A/V238A and K237_V238del do not affect protein folding. (**A**) Circular dichroism spectra of a2NT wild-type (red), mutant K237A/V238A (black), and mutant K237_V238.del (blue) in 50 mM Tris pH8.0 in the absence and (**B**) in the presence of 10mM SDS.

**Figure 3 ijms-25-02170-f003:**
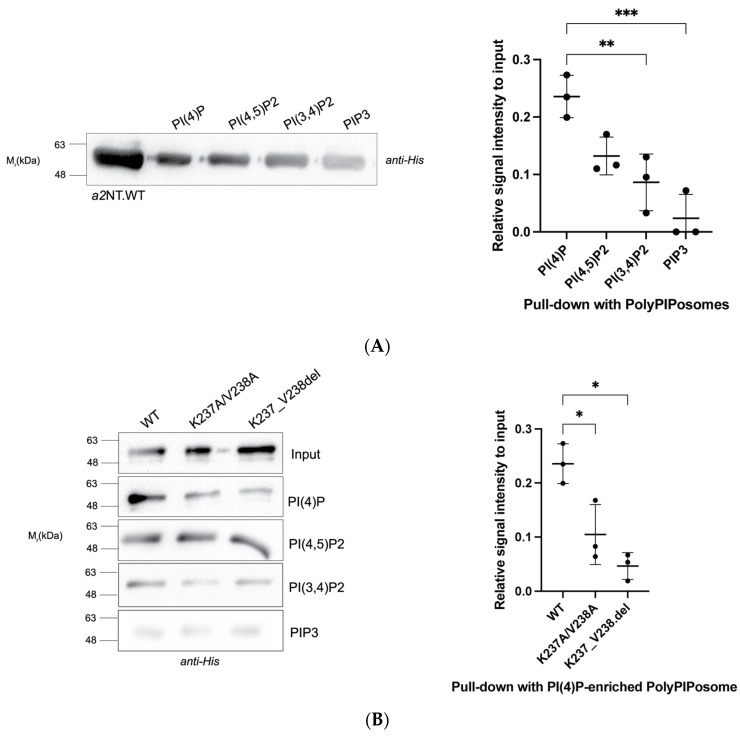
Mutations within the putative K^237^VKK^240^ binding motif reduced interaction of a2NT with PI(4)P-enriched liposomes in vitro. (**A**) Liposome pull-down assay with PolyPIPosomes (Echelon) enriched with the indicated PIPs and HIS-tagged a2WT. In total, 20 μg of protein was incubated for 1 h at room temperature with 20 μL of 1 mM PolyPIPosomes containing 5% of the indicated PIPs in binding buffer (50 mM Tris, pH 7.5, 150 mM NaCl, 0.05% Nonidet P-40). Additionally, 5 μg of purified HIS-tagged a2WT was used as the input for loading control. (Right) Quantification was performed by measuring the intensity ratio of protein pulled down with the liposomes relative to input (n = 3). (**B**) Liposome pull-down assay of WT and mutant proteins with PolyPIPosomes (Echelon) enriched with indicated PIPs (PI(4)P, PI(4,5)P_2_, PI(3,4)P_2_, and PIP3. (Right) Quantification by intensity ratio of WT and mutants pulled down with PI(4)P-enriched liposomes with respect to input. (**C**) Quantification by intensity ratio of WT and mutants pulled down with liposomes enriched with PI(4,5)P_2_, PI(3,4)P_2_, and PIP_3_ with respect to input. n = 3 for all figures. Error bars indicate ± S.D. Statistical significance was analyzed by one-way ANOVA with Dunnett’s multiple comparisons test comparing mutants to WT. * indicates *p* < 0.05. ** indicates *p* < 0.01, *** indicates *p* < 0.001.

**Figure 4 ijms-25-02170-f004:**
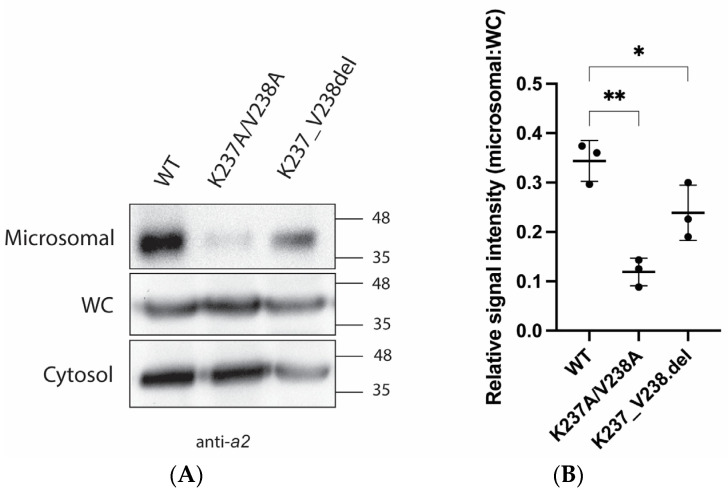
a2NT-WT co-purified with microsomal membranes, and mutants a2NT-K237A/V238A and K237_V238.del reduced membrane retention. (**A**) Plasmids containing a2NT wildtype (WT) and mutants K237A/V238A, K237_V238.del were transfected into HEK293 cells. Cellular fractionation performed to obtain cytosolic (cytosol) and microsomal fractions (microsomal). (**B**) Quantification was assessed by comparing the relative pixel intensity of microsomal fraction to whole cell extracts. (n = 3). Statistical significance was analyzed by one-way ANOVA with Dunnett’s multiple comparisons test comparing mutants to WT. * indicates *p* < 0.05, ** indicates *p* < 0.01.

**Figure 5 ijms-25-02170-f005:**
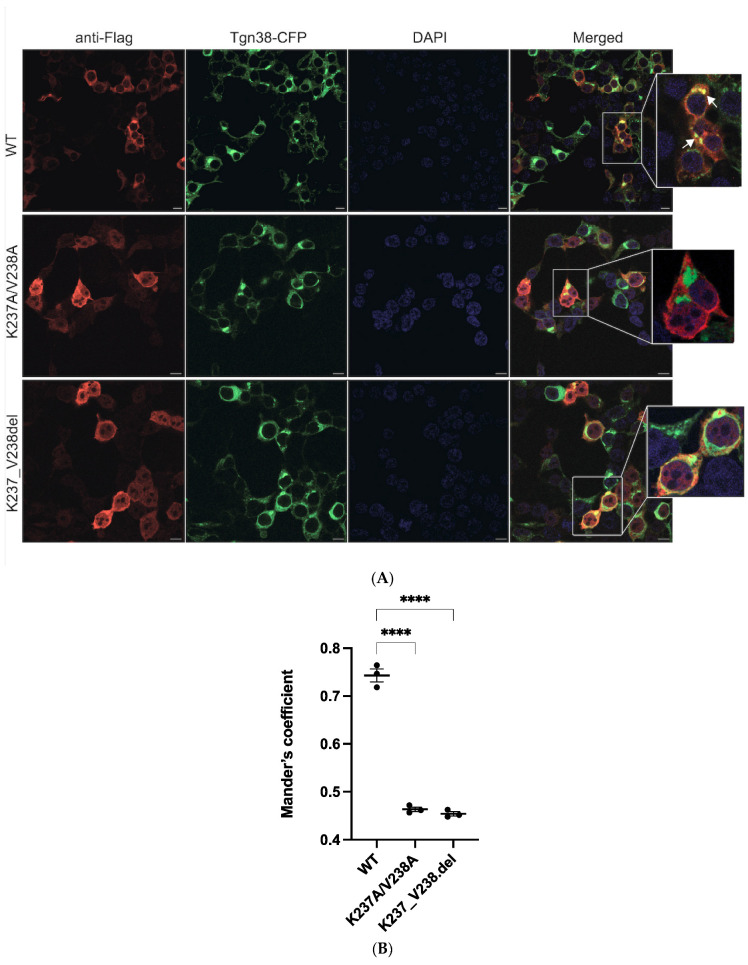
Mutations affect a2NT recruitment to Golgi. White arrows indicate the presence of a2NT at the Golgi. (**A**) Plasmids containing FLAG-tagged a2NT wildtype (WT), mutants K237A/V238A and K237_V238del were co-transfected with Tgn38-CFP in HEK293 cells. Cells were fixed, permeabilized, and stained for FLAG-tagged proteins (red) and DAPI (blue). (**B**) Quantification. A minimum of 30 cells from each cell line were measured for the intensity of the red signal at the vicinity of the Golgi membrane (green). Data represent mean value ± SEM from three independent experiments. Statistical significance was analyzed by one-way ANOVA with Dunnett’s multiple comparisons test comparing mutants to WT. **** indicates *p* < 0.0001, scare bar = 10 μm.

**Figure 6 ijms-25-02170-f006:**
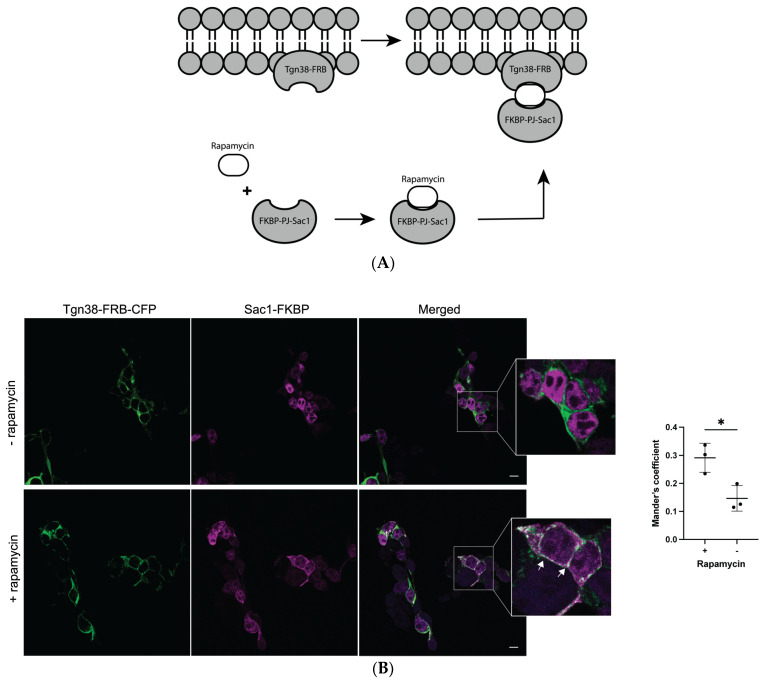
Rapamycin recruits Sac1-FKBP to the Golgi in the presence Tgn38-FRB. (**A**) Schematic illustration of rapamycin-induced dimerization. Sac1 phosphatase coupled to FK506 binding protein FKBP (Sac1-FKBP) and Golgi membrane anchor Tgn38 coupled to FKBP-rapamycin binding domain FRB (Tgn38-FRB-CFP) were dimerized by the addition of rapamycin. (**B**) Sac1-FKBP (magenta) recruitment to Golgi (white arrow), labeled by Tgn38 (green), upon treatment with 100 nM of rapamycin for 15 min at room temperature before fixing. Quantification: A minimum of 30 cells from each cell line. Data represent mean value SEM from three independent experiments. A paired *t*-test was run to analyze the significance in mean difference. * indicates *p* < 0.05, scare bar = 10 μm.

**Figure 7 ijms-25-02170-f007:**
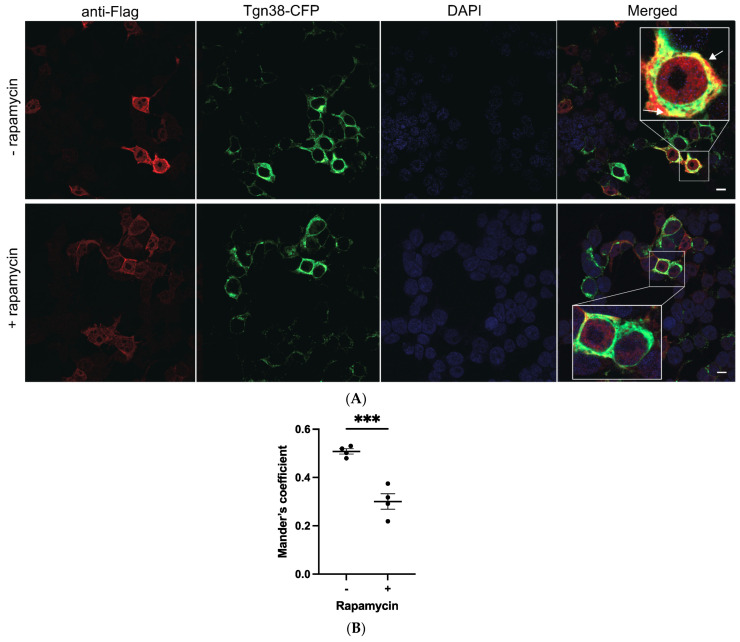
Depletion of Golgi PI(4)P impairs a2NT recruitment to Golgi. White arrows indicate the localization of a2NT at Golgi (**A**) Plasmids containing FLAG-tagged a2NT wildtype (WT) were co-transfected with Tgn38-FRB-CFP (green) and Sac1-FKBP in HEK293 cells. Cells were treated with 100 nM of rapamycin for 15 min at 30 h post-transfection. Cells were then fixed, permeabilized, and stained for FLAG-tagged proteins (red) and DAPI (blue). (**B**) Quantification was performed with a minimum of 30 cells from each cell line measured for the intensity of the red signal in the vicinity of Golgi (green). Data represent mean value SEM from three independent experiments. A paired *t*-test was run to analyze the significance in mean difference. *** indicates *p* < 0.001. scare bar = 10 μm.

## Data Availability

The original contributions presented in the study are included in the article material, further inquiries can be directed to the corresponding author.

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
