# Peer review of "The Human Mutation K237_V238del in a Putative Lipid Binding Motif within the V-ATPase a2 Isoform Suggests a Molecular Mechanism Underlying Cutis Laxa"

_ijms, 2024, doi:10.3390/ijms25042170_

Round 1
Reviewer 1 Report
Comments and Suggestions for Authors
The manuscript presented by authors showed that mutant K237A/V238A or K237_V238del in lipid binding motif within the V-ATPase a2 isoform prevented the interaction between a2NT and Golgi-enriched PIPs resulting in cutis laxa.
The current results presented were insufficient to support authors’ viewpoint. Firstly, the full and accurate materials haven’t been presented in manuscript and the experimental data was relatively weak. In addition, the quality of the supporting data was poor.
Apart from the specific issues in results, authors' intention was to elucidate the molecular mechanisms related to cutis laxa due to a2 mutations. However, authors did not elaborate on the specific molecular signal transduction of cutis laxa caused by a2 mutations. Moreover, all experimental explorations were performed in HEK293 cells, without involving in skin-related cells and lacking relevant data. Therefore, authors were unable to effectively associate experimental results with cutis laxa, and the research significance was not reflected in results. Overall, the content lacks certain novelty, making it hard to attract the interest of readers.
Comments on the Quality of English LanguageMinor editing of English language required.
Author Response
Dear Reviewer,
The material and method is now updated based on other reviewers' comments. Data presented in this study will nicely complement our previous publication about the a4 isoform (Chu A, Yao Y, Saffi GT, Chung JH, Botelho RJ, Glibowicka M, Deber CM, Manolson MF. Characterization of a PIP Binding Site in the N-Terminal Domain of V-ATPase a4 and Its Role in Plasma Membrane Association. Int J Mol Sci. 2023 Mar 2;24(5):4867. doi: 10.3390/ijms24054867. PMID: 36902293; PMCID: PMC10002524.)
The a2 K237_V238del mutation has been identified in patients with cutis laxa [48]. Here we show that this mutation disrupted PI(4)P interaction and compromised Golgi localization, suggesting a possible V-APTases-related mechanism underlying the disease.
Reviewer 2 Report
Comments and Suggestions for Authors
The manuscript entitled " The human mutation K237_V238del in a putative lipid binding motif within the V-ATPase a2 isoform suggests a molecular mechanism underlying cutis laxa" describes the generation of N-terminal (NT) domain of a2 isoform (a lipid-binding domain) of V- ATPases and mutations in this domain (a conserved basic motif) to understand their localization and regulation invivo. Further, authors have demonstrated the preferential binding of a2 NT with PI(4)P, a Golgi-enriched PIP, and show that PI(4)P at the Golgi helps retain the cytosolic N-terminal domain of a2 at the Golgi membrane. Additionally, authors have demonstrated the co-localization of a2NT with Golgi markers in the presence/ absence of Rapamycin. from the confocal analysis it was understood that Rapamycin depletion of PI(4)P diminished a2NT-Golgi localization. Thereby they conclude that a2NT is sufficient for Golgi retention, suggesting the lipid-binding motif is involved in V-ATPase targeting and/or retention. Mutational analyses suggest a molecular mechanism underlying how a2 mutations result in cutis laxa.
The manuscript can be accepted after minor revision.
1. I kindly request authors the provide the plasmid construct design (both bacterial and mammalian cell expression system) and the sequence results for the K237_V238del in a putative lipid binding motif within the V-ATPase a2 isoform.
2. Please include the mass spectrometry characterization for WT, mutation, and deletion of K237_V238del.
3. please verify the channel name of the fluorophore tagged Tgn38. is it a GFP tag or a CFP tag?
4. line 89, the multiple sequence figure 1, in the current draft should be modified. This figure was almost similar to their previous work.
Chu A, Yao Y, Saffi GT, Chung JH, Botelho RJ, Glibowicka M, Deber CM, Manolson MF. Characterization of a PIP Binding Site in the N-Terminal Domain of V-ATPase a4 and Its Role in Plasma Membrane Association. Int J Mol Sci. 2023 Mar 2;24(5):4867. doi: 10.3390/ijms24054867. PMID: 36902293; PMCID: PMC10002524.
5. What happens to PI4P interaction with a2 isoform expressing in cells expression deletion mutation of K237_V238del + supplemented with WT construct? Does it show a positive interaction with PI4P ?
6. Figure 4. Please include the appropriate loading control to support the cell fractionation. Also please include Molwt and epitope probed in the western blot analysis.
7. In Figures 5, and 7 DAPI channel is not visible please update the figures with better resolution.
8. For the biochemical assay the purity of the protein is important. Authors have performed only Ni-NTA purification and then Dialysis. Hence, I wonder about the purity levels of the a2 isoform in the present work. Please perform and provide the FPLC for the WT, mutation, and deletion mutations. Also the SDS-PAGE image to show the purity levels of the protein of interest.
9. binding kinetics for a2 NT and PI4P was not performed. it would be interesting to see the effect of different conc.of PI4P interaction with a2 isoform of V- ATPases
10. Please provide catalog and vendor details for antibodies and chemical reagents used in the current study
Reviewer 3 Report
Comments and Suggestions for Authors
In the manuscript "The human mutation K237_V238del in a putative lipid binding motif within the V-ATPase a2 isoform suggests a molecular mechanism underlying cutis laxa" an in vitro study of portions of the a2-subunit of V-ATPase (ATP6V0a2) has been conducted to test the hypothesis that mutations detected in the a2-subusti cutis laxa are involved in phosphoinositide binding. Previous work had suggested that a-subunit may interact with phosphoinositide's in the a4-subunit. Similar sites were detected in all four a-subunit isoforms, and it was noted that mutations in this binding site in a2 were found in patients with cutis laxa. Evidence was presnted that N-terminal constructs of a2 could be made with the relevent mutations without changing the 3D structure. As expected, the wild type constructs bound to various phosphoinositide's; the mutations reduced binding to PI4P. Mutant constructs when expressed in cells had reduced localization with isolated microsomes, Different location was confirmed by immunolocalization in HEK293 cells. In cells, depletion of PI4P altered the localization of the wild type construct in HEK cells. In general, this study shows solid evidence for the N-terminal of a2 binding to PI4P and that the mutations found in cutis laxa disrupt this interaction. I have a few minor suggestions.
1. Fig 5,. Tgn38-CFP is not mentioned in legend.
2. In experiments in 5-7, it is not clear whether transfection efficiencies and expression is similar for all constructs. If they are dissimilar then changes in distribution could be simply because binding sites are all covered, in all cases but with high expression relatively more of the construct is not able to bind its binding site.
Although the overexpression studies of a soluble region of a transmembrane protein, that is also part of a very complex larger enzyme are inherently complex, and difficult to interpret, the present study makes a strong case for the importance of the PI4P binding site.
Author Response
- Fig 5,. Tgn38-CFP is not mentioned in legend.
Updated legend: "Plasmids containing FLAG-tagged a2NT wildtype (WT), mutants K237A/V238A and K237_V238del were co-transfected with Tgn38-CFP in HEK293 cells."
2. In experiments in 5-7, it is not clear whether transfection efficiencies and expression is similar for all constructs. If they are dissimilar then changes in distribution could be simply because binding sites are all covered, in all cases but with high expression relatively more of the construct is not able to bind its binding site.
Expression levels has been checked to confirm equal level of expression.
Round 2
Reviewer 1 Report
Comments and Suggestions for Authors
After the revision, the material and method are improved. The experimental data is still too weak to support the present conclusions.
Comments on the Quality of English LanguageMinor editing of English language required.